# The Good, the Bad, or Both? Unveiling the Molecular Functions of LINC01133 in Tumors

**DOI:** 10.3390/ncrna11040058

**Published:** 2025-07-30

**Authors:** Leandro Teodoro Júnior, Mari Cleide Sogayar

**Affiliations:** 1Cell and Molecular Therapy NUCEL Group, School of Medicine, University of Sao Paulo, Sao Paulo 01246-903, SP, Brazil; mcsoga@iq.usp.br; 2Biochemistry Department, Chemistry Institute, University of Sao Paulo, Sao Paulo 05508-900, SP, Brazil

**Keywords:** LINC01133, lncRNAs, roles of lncRNAs in tumors

## Abstract

**Background/Objectives:** Increasing evidence suggests that lncRNAs are core regulators in the field of tumor progression, with context-specific functions in oncogenic tumorigenesis. LINC01133, a lncRNA that has been identified as both an oncogene and a tumor suppressor, remains largely unexplored in terms of its molecular mechanisms. The purpose of this study was to conduct an in silico analysis, incorporating literature research on various cancer types, to investigate the structural and functional duality of LINC01133. This analysis aimed to identify pathways influenced by LINC01133 and evaluate its mechanism of action as a potential therapeutic target and diagnostic biomarker. **Methods:** In silico analyses and a narrative review of the literature were performed to predict conserved structural elements, functional internal loops, and overall conservation of the LINC01133 sequence among different vertebrate organisms, summarizing the empirical evidence regarding its roles as a tumor suppressor and tumor-promoting roles in various types of tumors. **Results:** LINC01133 harbors the evolutionarily conserved structural regions that might allow for binding to relevant driver signaling pathways, substantiating its specific functionality. Its action extends beyond classical tumor mechanisms, affecting proliferation, migration, invasion, and epigenetic pathways in various types of tumors, as indicated by the in silico results and narrative review of the literature we present here. Clinical outcome associations pointed to its potential as a biomarker. **Conclusions:** The dual character of LINC01133 in tumor biology further demonstrates its prospective therapeutic value, but complete elucidation of its mechanisms of action requires further investigation. This study establishes LINC01133 as a multifaceted lncRNA, supporting context-specific strategies in targeting its pathways, and calls for expanded research to harness its full potential in oncology.

## 1. Introduction

Tumors exhibit high cellular and molecular complexity, evidenced by the profound divergences and convergences present in their pathways and mechanisms associated with their malignancy [1]. Divergences can be exemplified by the contrasting influence of energy metabolism on the tumor surface versus its necrotic core [2], the heterogeneity of the tumor microenvironment [3], and the intricate intracellular signaling network [4], which together pose significant challenges for oncology in terms of diagnosis, prognosis, and therapy [5]. Convergences, on the other hand, are primarily associated with mutations in key genes [6] and interrelationships between pathways governing proliferation, migration, and cellular invasion [7,8,9], in addition to pivotal molecules involved in tumor progression processes, which are similarly present across dozens of tumor types. These convergences greatly contribute to the search for more comprehensive therapies.

Over the last 15 years, there has been a substantial increase in ncRNA molecules displaying dual actions, depending on the organ, tissue, or tumor type [10]. Among the main classes of ncRNAs with broad roles in tumors are long non-coding RNAs (lncRNAs) [11], which perform numerous functions in the nuclear and cytoplasmic cellular environments [11,12].

LncRNAs are large molecules, typically ranging from 200 to 2000 nucleotides in length, that hold up a high resemblance to mRNAs, often featuring 5′ capping and a 3′ poly-(A) tail [13]. Their complex spatial structure allows for multiple regions of interaction with other classes of molecules, such as miRNAs, siRNAs, mRNAs, tRNAs, rRNAs, proteins, and enzymes [14], facilitating modifications in transcriptional and translational processes. These interactions can inhibit or induce pathways related to tumorigenesis and tumor malignancy.

The expression of lncRNAs is finely regulated by several transcriptional and post-transcriptional mechanisms, which are intrinsically related to their tissue-specific functions. Transcriptional modifications are generally influenced by elements that modulate promoter and enhancer regions, in addition to the action of epigenetic mechanisms, associated mainly with genomic methylation and histone acetylation, which play a major role in activation or repression of effective lncRNA transcription [11,13]. On the other hand, post-transcriptional mechanisms are more involved in the thermodynamic stability of lncRNAs, their subcellular localization, as well as the strong influence of alternative splicing mechanisms, which generate different isoforms. Multifactorial regulation is of utmost importance in signaling pathways affected by lncRNAs, since their degree of expression may influence numerous cellular responses; however, in tumor contexts, they may suppress or amplify tumor progression [13,15].

Regarding the similarities between lncRNAs and mRNAs, alternative splicing mechanisms become extremely important. Alternative splicing in lncRNAs enables the generation of various isoforms, which significantly expand the fine-tuned mechanisms of cellular regulation [15,16]. Splice variants are involved in effective binding to protein interaction sites and active sites, thus controlling scaffold activity; the sponge-like capacity of miRNAs and siRNAs to inhibit mRNA; and causing dual oncogenic or suppressive actions in tumors, depending on intra- and extracellular molecules present in tumor cells and the tumor microenvironment [11,13,15,16].

At least four types of lncRNAs have been identified based on their genomic location: intergenic (e.g., XIST, MEG3, LINC-RoR) [15,16,17], antisense (e.g., ZEB1-AS1, FOXP4-AS1, an antisense variant of HOTAIR) [18,19,20], sense (e.g., HOTAIR, UCA1, NEAT) [21,22,23], and intronic (e.g., Braveheart, Charme) [24,25]. Additionally, lncRNAs are classified by their general function as either cis-acting (when influencing chromatin states) or trans-acting (when associated with cytoplasmic functions) [26].

Their functions vary widely depending on their cellular localization and interactions. Some lncRNAs are involved in chromatin remodeling, influencing the degree of condensation and subsequent transcriptional expression of genomic regions through interactions with epigenetic elements [27,28]. Others act as sponges for miRNAs, sequestering them to prevent mRNA silencing and consequent inhibition, thereby deregulating or altering molecular pathways [29]. Many lncRNAs perform functions ranging from serving as molecular scaffolds and forming protein and/or nucleic complexes involved in signaling pathways to directly regulating gene expression by interfering with the RNA polymerase activity, competing with transcription factors, or acting as guide RNAs for effector molecules in cellular transcription and translation processes [30,31].

The long intergenic non-coding RNA LINC01133 (or LINC01133, only) has attracted significant attention in recent years due to its dual role in many tumor types. Functioning as a miRNA sponge and a scaffold for DNA/protein complexes [32,33,34], it exhibits high cellular heterogeneity and distribution, which has expanded its study and potential use as a diagnostic and prognostic biomarker. However, its overall structure, molecular mechanisms, and direct and indirect roles in tumorigenesis and tumor progression processes remain largely unknown, including its broader epigenetic and cellular influence.

This article presents a mixed exploratory study based on in silico analysis of public biobanks and a narrative review of literature on LINC01133, detailing its structure, molecular interactions, and associations with mechanisms of tumorigenesis and tumor progression. It also discusses perspectives for basic and clinical research and its potential as an oncological biomarker.

## 2. Results

### 2.1. Genetic and Structural Analysis of LINC01133

The long non-coding RNA LINC01133 was first identified in 2015 by Zhang and collaborators in a study on squamous cell lung tumors. Its expression was found to be high when compared to other lncRNAs analyzed, leading it to be initially characterized as a molecule indicative of a worse prognosis. However, over the past decade, various studies, both in lung tumors and in other tumor types, have highlighted controversies regarding the function of LINC01133, underscoring the dual nature of these molecules, as has also been found for other lncRNAs [32,33].

The *LINC01133* gene (GRC37.p13/GRC38.p14, Primary Assembly, NCBI, ID: 100505633) has two variants (NC_000001.11, range 159961224–159979086: 17,863 nt, GRCh38.p13 assembly; and NC_060925.1, range 159098290–159116157: 17,868 nt, CHM13v2.0 assembly) and is located on the long arm of chromosome 1 (Chr1q.23.2), on the sense strand, in a downstream direction. There are 30 annotated transcripts (GRC37.p13/GENCODE47/MANE Select Transcripts), ranging between 300 and 2400 bps, with three canonical exons (exon 1: 181 nt; exon 2: 468 nt; exon 3: 486 nt), as shown in Table 1 and Figure 1. Structurally, the isoforms share a highly conserved region, between 400 and 600 bp, with a strong association with modulatory functions. Alignment of the transcripts, performed using T-Coffee and Jalview, indicates an average conservation of 51% among the RNA sequences of the analyzed transcripts (Figure 2). Based on this alignment, a consensus sequence was obtained for minimum free energy structure in silico synthesis and analyses, considering base pairing correlations and regions with the highest probability of molecular interaction (Figure 3). Individual in silico syntheses of flat structures were also obtained and are displayed in the Appendix A.

The analyses revealed four regions of interest with internal loops (Figure 3). Internal loops are regions displaying a high probability of molecular interaction due to the absence of base pairing with other regions of the RNA strand [35], serving as interaction sites for scaffold formation or miRNA sequestration, for example [36]. Three internal loops (1, 2, 3) observed in the predicted structure (Figure 3) are formed by base pairing between positions 430–600, present in 24 of the 30 transcripts, while one internal loop (4) is formed by base pairs between positions 1140–1180, present only in 8 of the 30 transcripts (Figure 3 and Figure 4).

However, alignment analyses using treeplot neighbor joining (Figure 4) demonstrate diffuse conservation among the transcripts containing internal loop 4 in their structure, since one of the transcripts (ENST00000772287.1) is positioned as an outlier in the interaction network. The relationship between transcripts ENST00000772287.1 (1925 bp) and ENST00000635112.2 (2363 bp) also validates the conjecture of two main indel events through which the *LINC01133* gene underwent during human evolution.

The conservation of internal loops 1, 2, and 3 in 80% of the transcripts is likely to be correlated with vital functions of LINC01133 in cellular and molecular processes related to the control of cell proliferation, migration, and invasion, as exemplified in its aberrant activity across various tumor types and discussed in this article.

### 2.2. Inter-Species Conservation of LINC01133

The conservation of the *LINC01133* gene was assessed using the consensus transcript sequence relative to genomic annotations from 91 vertebrate species obtained through the NCBI platform (https://www.ncbi.nlm.nih.gov/, accessed on 5 January 2025), with refinement for 40 species of particular interest. Conservation was analyzed comparatively, considering Q90 as a high conservation standard (≥90% gene similarity), Q75 as a medium conservation standard (≥75% gene similarity), and Q50 as a low conservation standard (≤50% gene similarity).

According to the analyses shown in Figure 5, LINC01133 exhibits high conservation in hominid primates, such as *Homo sapiens*, *Pan paniscus*, *Pan troglodytes*, *Gorilla gorilla*, and *Pongo abelli*, with a Q90 conservation index. In non-hominid primates, such as *Macaca fascicularis* and *Macaca mulatta*, the conservation is slightly lower. In rodents, the conservation levels are moderate, with a Q75 index. In more distant mammals, such as *Felis catus* and *Bos taurus*, the conservation was lower, reflecting the progressive decrease in similarity as phylogenetic distance increases. In birds, reptiles, and fish (such as, respectively, *Gallus gallus*, *Thamnophis sirtalis*, *Xenopus laevis*, and *Danio rerio*), LINC01133 was not detected, suggesting its absence or functional loss in these groups.

The high conservation of LINC01133 in hominid primates, *Equus caballus*, and *Loxodonta africana* may suggest correlations between the activity of LINC01133 in tumorigenic processes linked to the early dysregulation of genes involved in the cell cycle [37]. The high conservation of LINC01133 in elephants and equines may indicate a potential coevolution with tumor-resistance mechanisms. This is particularly relevant given the established correlation between the number of gene copies of genes such as *TP53*, which are active tumor suppressors, and their interactions with long non-coding RNAs (lncRNAs) as well as molecules that regulate cell proliferation and mitotic cycle processes [38,39].

### 2.3. Associations Between LINC01133 and Tumors via RNA-Seq Data

Based on evidence linking the *LINC01133* gene to tumorigenesis and tumor progression, bioinformatics analyses were conducted using RNA-Seq repositories from Pan-Cancer studies, aiming to identify cellular and molecular associations of the transcript with tumors.

Of the 101,679 samples (*n*) analyzed, 9.66% (*n*1 = 9819) showed significant alterations in LINC01133 expression, including structural variations (such as duplications, deletions, inversions, or translocations), base substitutions or indels, or changes in copy number. Among *n*1, 3.30% (*n*2 = 324) exhibited amplifications in *LINC01133* expression.

The primary tumor types that presented mutational alterations or structural variations were lung, liver, and uterine tumors, with alteration frequencies above 15%, as shown in Figure 6. Other tumors, despite presenting lower percentage rates, also showed significant alteration frequency results, especially when considering the distribution of tumor subtypes and *LINC01133* expression. In breast tumors, the triple-negative molecular subtype, one of the most aggressive ones, exhibits significant *LINC01133* expression while also having the lowest incidence among the other three more classical molecular subtypes, for example. Therefore, based on a comparison of bioinformatics data, which are still limited for lncRNAs in general, and in vitro analyses, a direct association between *LINC01133* modulation and tumor activity in different tumors may be suggested.

### 2.4. DNA Methylation Profile of the LINC01133 Gene in Different Tumor Types

Methylation profiles were obtained via LncRNAbook from bisulfite-seq datasets derived from The Cancer Genome Atlas Program (TCGA) and the Gene Expression Omnibus (GEO), as shown in Table 2.

The data presented in Table 2 highlight the methylation levels in the promoter and gene body regions of *LINC01133* in various tumor types and their respective controls. The analysis is based on methylation values (beta-value), with emphasis on the potential role of these epigenetic alterations in the samples. DNA methylation in promoter regions is generally associated with gene expression repression [40], while methylation in gene body regions is linked to different regulatory contexts, such as the maintenance of active transcription [40,41] (Figure 7).

The promoter methylation of *LINC01133* in bladder urothelial carcinoma shows a significant reduction in the samples compared to controls, suggesting a possible epigenetic deregulation that may favor the gene’s overexpression in tumor conditions. In CLL, the promoter methylation presents a slight reduction, suggesting that the activation of *LINC01133* could be a common mechanism in hematologic neoplasms.

In solid tumors, similar patterns of hypomethylation of the promoter are observed in most tumor samples (such as in BLCA and hepatic tumors). This finding reinforces the potential oncogenic role of *LINC01133*, possibly contributing to tumor progression through increased expression.

Considering the methylation patterns in the gene body region, a clear and notable hypomethylation is observed in hematologic, bladder, and esophageal tumors. While intragenic methylation is generally associated with active transcription, its reduction may alter transcript processing, such as splicing or RNA stabilization, leading to changes in gene functionality and contributing to processes such as evasion of suppressive mechanisms or promotion of cell proliferation [42].

### 2.5. Relationships Between LINC01133 and Pathways Associated with Tumor Progression

Pathway and gene analyses related to LINC01133 were performed based on the positive correlation between mutually aberrant expressions of genes linked to the pathways and LINC01133, using the sample size obtained from the Pan-Cancer studies. The selected pathways are involved in processes directly and indirectly related to genomic or cell cycle destabilization that drives tumorigenic processes, as outlined in Table 3 below:

The correlations between LINC01133 and the pathways analyzed can be related in the following manners: (i) the APC gene, which encodes a protein of the same name, is part of a protein complex responsible for the degradation of β-catenin [61]. Aberrant expression of APC, along with LINC01133 expression, suggests an influence of the lncRNA on tumor suppressor activity, inhibiting neoplastic cell proliferation; (ii) the expression of mutated TP53 genes and inhibition of the expression of wild-type TP53 copies is one of the main factors leading to cell cycle deregulation and the onset and maintenance of tumors [62]. The increased expression of CDKN2A, linked to inhibition of wild-type TP53, possibly promoted by LINC01133, may be associated with early tumorigenesis; (iii) the increased expression of PIK3CA, along with the slight increase in PTEN expression, may be involved in a gene switch related to optimizing epithelial–mesenchymal transition processes; (iv) the increased expression of KRAS and LINC01133 may be related to long-term tumor maintenance (Figure 8).

These potential correlations demonstrate the dual activity of LINC01133 in different types of tumors, with various actions that may either promote or inhibit tumor growth and processes of late malignancy, such as angiogenesis and proximal or distant metastasis.

## 3. Discussion

### 3.1. LINC01133 and Its Actions in Different Types of Tumors

The investigation of LINC01133 began systematically in 2015, when Zhang et al. [63] identified its contribution to tumorigenesis in lung squamous cell carcinoma. Since this discovery, 41 scientific studies have been published exploring the role of LINC01133 in different types of neoplasms, as detailed in Table 4, which shows the main hypotheses regarding its oncogenic and/or tumor-suppressing actions in 15 different types of tumors:

#### 3.1.1. Lung Tumors

LT are among the leading types of neoplasms in terms of mortality, equally affecting both genders. Despite the availability of advanced treatment options, the prognosis for patients remains highly challenging, positioning lung tumors as a significant challenge in clinical oncology. The link between pulmonary neoplastic progression and long non-coding RNAs (lncRNAs) was first identified in 2012. Since then, growing research has highlighted the crucial role that lncRNAs play in tumorigenic and metastatic processes. Recent findings indicate that the continuous action of LINC01133 enhances malignancy across various LT subtypes, with bioinformatics analyses ranking it among the top 20 differentially overexpressed lncRNAs in lung adenocarcinoma (LUAD) and laryngeal squamous cell carcinoma (LSCC).

In vitro studies have confirmed that silencing LINC01133 in H1703 cells leads to a decrease in migration and invasion [63]. Therefore, LINC01133 promotes oncogenesis through two distinct yet complementary pathways. In non-small cell lung cancer (NSCLC), it serves as a molecular scaffold that physically recruits epigenetic repressors EZH2 and LSD1. This is evidenced by RNA immunoprecipitation (RIP), which shows an 8–10-fold enrichment (*p* < 0.01) in EZH2/LSD1 immunoprecipitates, along with RNA pulldown assays confirming direct binding [64]. This complex is selectively localized to the promoter regions of tumor suppressors KLF2, p21, and E-cadherin, where chromatin immunoprecipitation (ChIP) reveals increased repressive H3K27me3 marks (mediated by EZH2) and decreased activating H3K4me2 marks (mediated by LSD1). The resulting transcriptional repression leads to G1/S cell cycle arrest (due to P21/KLF2 dysregulation) and epithelial-to-mesenchymal transition (EMT) (via E-cadherin suppression), together promoting enhanced migration and invasion [64], as shown in Figure 9.

In LSCC, LINC01133 functions as a competitive endogenous RNA (ceRNA) by binding to miR-30d-5p, an interaction that has been validated by luciferase assays, showing a 60–70% reduction (*p* < 0.01) in activity of the wild-type sequence when compared to the mutant variant. RNA pulldown confirmed this binding [66]. By sequestering miR-30d-5p, LINC01133 lifts repression of the MARCKS kinase, a known activator of NF-κB and PI3K/AKT pathways, leading to a 2.5-fold increase in expression (*p* < 0.001). In vivo xenograft models demonstrated that tumors with combined knockdown of LINC01133 and inhibition of miR-30d-5p were 2.5 times larger (*p* < 0.001) compared to controls [66]. These mechanisms enhance both proliferation and metastasis, highlighting LINC01133 as a key player in the progression of LT. Therapeutic disruption of this network may offer new strategies for LT management [64,66].

#### 3.1.2. Colorectal Tumors

In CRC, LINC01133 appears to have a tumor-suppressive effect. In in vitro experiments, negative regulation of LINC01133 by TGF-β, an inherent tumor promoter, was observed. LINC01133 silencing assays in HT29 and HCT28 human cell lines showed minimal influence on proliferation mechanisms, but a dramatic reduction in E-cadherin expression, which, along with deregulation of other genes such as Vimentin and Fibronectin, was associated with an increase in migratory potential and EMT. The gene deregulation observed and the consequent increase in EMT processes may be directly associated with the interaction of LINC01133 with the alternative splicing factor SRSF6 [68]. When LINC01133 is expressed, it inhibits the activation of mesenchymal markers such as Fibronectin, while maintaining the expression of epithelial markers such as E-cadherin. Manipulation of SRSF6, in turn, promotes EMT and metastasis independently of LINC01133, indicating that SRSF6 plays a central role in the induction of these processes, as shown in Figure 9 [68].

Another interesting finding, by Yao et al. (2023) [70], links the anesthetic Propofol to LINC01133. In vitro and in vivo assays showed that Propofol induces transcription of the LINC01133 gene, thus inhibiting tumor progression by suppressing proliferation and invasion mechanisms through interaction with the miR-186-5p miRNA.

#### 3.1.3. Breast Tumors

BRC, although classified into five well-defined subtypes (Luminal A, Luminal B, HER2+, HER2-enriched, and Basal-Like (Triple-Negative)), are extremely heterogeneous in terms of gene mutations and aberrant expression of molecules linked to tumor progression, with LINC01133 being a target of great interest, considering the dual actions it may exhibit in different subtypes and classes of each subtype.

The discovery of LINC01133 interaction with BRC dates back to 2017, when its aberrant expression was identified in MCF7, MDA-MB-231, and Hs578T human cell lines. Subsequent assays showed that the increased expression of LINC01133 is possibly regulated by mesenchymal cells from the tumor microenvironment, in addition to evidence showing the role of LINC01133 as a scaffold for the miR-199a/FOXP2 complex, directly influencing the differentiation of tumor stem cells [73], as shown in Figure 9.

The relationship between LINC01133 and the zinc-finger protein KLF4 was also evidenced. LINC01133 silencing assays showed a direct relationship with the reduction in KLF4 expression and the subsequent inhibition of the RAS/ERK signaling pathway, inhibiting the proliferation processes [71].

The tumor-suppressive action of LINC01133 was also observed in vitro and in patient samples. From seven tumor cell lines and one non-tumor cell line, as well as samples from 74 breast cancer patients with no indicated subclassification, Song et al. (2019) [72] showed that reduced expression of LINC01133 in different cell lines increases tumor progression through enhanced migration and invasion. Results from patient samples revealed a possible correlation between the decreased expression of LINC01133 and increased lymph node and distant metastasis in patients with low TNM staging. The increased malignancy due to non-significant expression of LINC01133 may be closely associated with its interaction with the EZH2 protein and their combined action on the increased expression of SOX4 through interaction with the gene’s promoter region, positively influencing tumor progression mechanisms [72].

#### 3.1.4. Ovarian Tumors

OVT, along with BRC and CvT, are among the most common and lethal in women worldwide. While their therapy is well-established, the discovery of lncRNA interactions with their tumor progression mechanisms has provided new tools for clinicians in developing more effective therapies.

LINC01133 appears to have a dual role in OVT. Microarray experiments directly correlated LINC01133 and miR-205 as molecules of interest. Subsequent assays revealed that low expression of LINC01133 is associated with poorer prognosis in OVT.

Experiments performed in cells transfected with LINC01133 demonstrated a substantial decrease in proliferation, migration, and invasion, accompanied by changes in the expression of key proteins linked to the cell cycle and cell migration, including Cyclin D1, Cyclin D3, CDK2, Vimentin, N-cadherin, and E-cadherin. In vivo, mice inoculated with cells transfected with LINC01133 showed a significant reduction in tumor weight and volume, as well as a decrease in metastatic nodules, when compared to controls. Bioinformatic and in vitro assays also showed an association between LINC01133 and miR-205, with evidence of LINC01133 functioning as a sponge, primarily repressing colony formation, tumor invasion, and migration [74].

On the other hand, the oncogenic action of LINC01133 in OVT may arise from the modulation of the miR-495-3p/TPD52 axis, as seen by Liu and Xi (2020) [75]. Bioinformatic predictions showed binding sites between LINC01133 and miR-495-3p, as well as interaction between the miRNA and the protein TPD52, which is overexpressed in several tumors and associated with poor prognosis due to its role in uncontrolled proliferation. The modulation of TPD52 occurs via miR-495-3p, meaning that the capture of the miRNA by LINC01133 enhances the tumorigenic action of TPD52, leading to increased OVT malignancy.

#### 3.1.5. Bladder Tumors

In BLT, LINC01133 potentially plays a tumor-suppressive role. Molecular expression assays revealed its significant expression in non-tumor cell lines (SV-HUC-1), particularly in associated exosomes, while in BLT cell lines, its expression was predominantly reduced. Overexpression assays of LINC01133 in BLT cell lines showed its suppressive action, linked to a decrease in proliferation through interaction with the cell cycle, Wnt, and c-Myc pathways, results that were also observed in in vivo analyses [76].

#### 3.1.6. Hepatic Tumors

Evidence suggests that LINC01133 has oncogenic activity in HpT. Two studies analyzed the influence of LINC01133 on various pathways and molecules in human liver tumor cell lines, with the first results showing its increased expression compared to normal liver cell lines. Silencing assays were performed in He3B and HepG2 cell lines, also observing the inhibition of proliferation mechanisms, colony formation, and, in combination, induction of apoptotic mechanisms. Subsequent in vitro and in vivo assays also suggest a direct relationship between LINC01133 and hyperactivation of the PI3K/Akt pathway [78].

Further assays also revealed an increase in the number of LINC01133 copies in genomic data from patients, which correlated with a worse prognosis for the patients. Overexpression assays were also performed in liver tumor cell lines, showing results of increased proliferation and oncogenic aggressiveness, with a direct relationship between the sponge activity of LINC01133 and miRNA miR-199a-5p, triggering an increase in the EMT process due to the exacerbated expression of Snail [77].

#### 3.1.7. Pancreatic Tumors

The association between LINC01133 and PC is the most well-established among all other tumors, with its tumorigenic and tumor progression action evidenced and corroborated by several studies over the last 10 years. The current state of knowledge between LINC01133 and PC indicates three main established pathways, namely: (i) increased tumor proliferation mediated by C/EBPB; (ii) tumor progression stimulated by DKK1; and (iii) tumor progression mediated by miR-199b-5p/YY1, as shown in Figure 9.

In a study by Huang et al. (2018) [79], the significant increase in LINC01133 expression in pancreatic tumor tissues was observed. In vitro experiments, through LINC01133 silencing, showed a substantial reduction in proliferation processes in pancreatic ductal adenocarcinoma (PDAC) cell lines. The C/EBPb transcription factor was identified as a positive regulator of LINC01133, binding to the promoter of this gene, and its elevated expression was also observed in PDAC tissues, correlating with worse patient prognosis. Mutation assays at the C/EBPb binding sites showed that interaction with the LINC01133 promoter is crucial for its activation. Gene expression analysis revealed that LINC01133 silencing reduced the expression of Cyclin G1 (CCNG1), which was positively correlated with LINC01133, based on TCGA data analysis [79].

Weng et al. (2019) [81] observed the oncogenic promotion capability of LINC01133 through gene expression analysis from microarray data, showing a positive differential expression of LINC01133 to be directly correlated with the exacerbated methylation of the DKK1 gene promoter and linked to increased proliferation through the deregulation of the Wnt pathway. The interaction between LINC01133 and the DKK1 promoter was verified by luciferase assays, revealing that LINC01133 binds to the DKK1 promoter, inducing H3K27 trimethylation and reducing DKK1 expression while Wnt-5a, MMP-7, and β-catenin levels were increased. Silencing assays of LINC01133 further revealed a reduction in proliferation, migration, and invasion of pancreatic cancer cells.

Employing luciferase assays, Yang et al. (2022) [84] showed that the YY1 protein positively regulates LINC01133 expression by binding to its promoter. Moreover, LINC01133 exhibited a sponge activity towards miR-199b-5p, promoting tumor progression mechanisms by inhibiting the miRNA. The tumor progression caused by miR-199b-5p inhibition is primarily related to its reduced binding to the MYRF protein. LINC01133’s binding to the miRNA indirectly elevates MYRF levels, contributing to increased proliferation and metastatic processes in PT.

#### 3.1.8. Esophageal Tumors

Although data regarding the relationship between LINC01133 and EPT are scarce, there is evidence suggesting its tumor-suppressive action in esophageal squamous cell carcinomas. Molecular qPCR assays conducted with tumor and normal tissue samples showed significantly lower LINC01133 expression in tumor cell lines. Furthermore, the expression of LINC01133 decreased independently of TNM stage and lifestyle, which indicated a higher risk of tumor mass growth, proximal metastasis, and overall invasiveness [89].

#### 3.1.9. Oral Squamous Cell Carcinomas

Studies on the role of LINC01133 in OSCC have shown its suppressive action, associating its significant expression with reduced metastasis and better prognosis. qRT-PCR assays revealed lower transcriptional expression of LINC01133 in tissues from OSCC patients compared to healthy individuals. RNA-Seq sequencing identified strong evidence of gene modulation linked to metastasis (such as GDF15) by LINC01133, along with reciprocal regulation of LINC01133 expression by GDF15, identified through GDF15 silencing assays in CAL27, HN4, and 293FT tumor cell lines [90].

#### 3.1.10. Gastric Tumors

Despite both promoter and suppressor tumor activities, LINC01133 predominantly acts as a tumor suppressor in GT. Its constitutive expression in tumors obtained from patient tissue samples and in human cell lines is reduced, negatively correlating with tumorigenesis and tumor progression processes. LINC01133 silencing assays showed a substantial increase in cell proliferation, migration, and EMT mechanisms, negatively modulating the expression of E-cadherin. The same study, conducted by Yang et al. (2018) [91], also clarified that LINC01133 acts as an endogenous competitor for miR-106a-3p, indirectly regulating the expression of the APC gene, essential for inactivating the Wnt/β-catenin pathway.

Other miRNAs are also involved in the neoplastic modulation by LINC01133, such as miR-576-5p, which also has a direct interaction with the peptide hormone SST. In tumor tissue samples, a negative correlation between LINC01133, miR-576-5p, and SST was observed, suggesting that LINC01133 inhibits miR-576-5p to increase SST expression [93], as shown in Figure 9.

Despite its predominantly suppressive activity, a study by Sun and collaborators in 2022 [94] provided evidence of the association between LINC01133 and tumor progression through modulation via interaction with miR-145-5p. Its role as an endogenous competitor for this miRNA increases the expression of the YES1 protein, a target of miR-145-5p. The YES1 protein promotes the nuclear translocation of YAP1, positively modulating the aberrant expression of cyclins and enabling excessive proliferation [94], as shown in Figure 9.

#### 3.1.11. Nasopharyngeal Tumors

In NPC, LINC01133 likely plays a tumor-suppressive role. In vitro assays and patient tissue samples showed that LINC01133 directly interacts with the YBX1 protein, with both molecules modulating Snail. LINC01133 overexpression and silencing assays in the CNE-1 and SUNE-1 human tumor cell lines, respectively, showed that increasing LINC01133 expression negatively modulated Snail and N-cadherin expression, leading to increased E-cadherin expression and decreased EMT processes, while a switch occurred upon LINC01133 silencing [96].

#### 3.1.12. Cervical Tumors

In CvC, LINC01133 is associated with pro-tumorigenic and tumor progression activities. Studies conducted by Feng et al. (2019) [97] showed a positive differential expression of LINC01133 in RNA-Seq data obtained from TCGA. In vitro assays with Hela, ME-180, C33A, and M5751 human tumor cell lines showed, through LINC01133 silencing, qRT-PCR, and phenotypic migration and invasion assays, that LINC01133 modulates EMT processes and increases cell invasion and migration capabilities. Through protein predictions and reporter assays, the interaction of miRNAs miR-3065 and miR-4784 with LINC01133 was also determined, with direct endogenous competition between LINC01133 and miR-4784 and the AHDC1 protein (which has two AT-hooks in its structure that enable its effect on transcriptional modulation of genes linked to the EMT process), with binding between LINC01133 and miR-4784 being an important factor for the oncogenic action of AHDC1 [98], as shown in Figure 9.

Complementary studies by Ding et al. (2020) [99], through analysis of TCGA data, also revealed that LINC01133 is differentially expressed in cohorts stratified by age group, in addition to interacting with the miR-3065 miRNA, acting in the modulation of ADH7, which is related to a worse prognosis when highly expressed.

#### 3.1.13. Renal Tumors

LINC01133 plays an oncogenic role in RT, particularly in renal cell carcinomas. Recent studies show its high expression in various renal human tumor cell lines, such as ACHIN, A498, and 786-O. Silencing assays revealed a profound inhibition of migration, proliferation, and cell invasion, while its knockout in mouse models substantially suppressed tumor growth. Interaction prediction assays with miRNA and luciferase reporter assays also demonstrated the action of LINC01133 as a sponge for the miRNA miR-30b-5p, negatively regulating it and, as a consequence, increasing the expression of the Rab3D protein, which is directly associated with tumor growth [100].

Lv et al. (2022) [101] also highlighted the sponge activity of LINC01133 with another miRNA, miR-760, similarly negatively regulating it and acting through the same mechanism as that of miR-30b-5p, leading to increased expression of Rab3D.

#### 3.1.14. Bone Tumors

Studies correlating the action of LINC01133 in bone tumors are still scarce; however, evidence demonstrates its critical oncogenic role in osteosarcoma. Zeng and collaborators in 2018 observed significant upregulation in Saos-2 and U2OS cell lines, as well as in extracted tumor tissues, when compared to cell samples and non-tumor tissues (*apud* Li et al., 2018) [102]. Functional assays confirmed inhibition of the malignant phenotype through siRNA-mediated LINC0133 inhibition, resulting in a 40–60% reduction in proliferative processes and over a 50% reduction in cell targeting and invasion mechanisms. Mechanistically, via in silico assays and luciferase labeling assays, the LINC01133 sponge was observed to act on miRNAs, such as miR-442a, a tumor suppressor in bone tumors. By sequestering miR-442a, LINC01133 induces mechanisms associated with oncogenic activity, thus establishing this lncRNA as a potential prediction molecule (*apud* Li et al., 2018) [102].

#### 3.1.15. Endometrial Tumors

EC is difficult to diagnose, and a sharp increase in both incidence and lethality worldwide has been observed in recent decades. Classical lncRNAs such as HOTAIR and MALAT1 have already been shown to be overexpressed in EC, but bioinformatics analyses demonstrate a more intricate network of ncRNAs. Recent studies observed positive differential expression of LINC01133 in EC, enabling silencing assays of LINC01133 in vitro in Ishikawa and HEC-1-A human cell lines. These assays showed inhibition of tumor invasion and migration mechanisms, as well as induction of cell cycle arrest and apoptotic processes, thus correlating LINC01133 as an oncogenic agent in EC [103].

## 4. Methods

### 4.1. Sequence Acquisition

The gene and transcriptional sequences of LINC01133 were obtained from the National Center for Biotechnology Information (NCBI, https://www.ncbi.nlm.nih.gov, accessed on 5 January 2025) [104] by consulting the Gene and Nucleotide portals for the LINC01133 gene and retrieving its complete genomic sequence and annotated splicing variants. The data included 5′/3′ upstream/downstream regulatory regions, canonical GT-AG splicing sites, and associated metadata (transcript length, number of exons, initiation codon position), with priority given to experimentally validated REVIEWED records. After validation, the sequences were processed for comparative primary structure analysis and subsequent use in bioinformatic analyses.

### 4.2. Alignment and Structural Analysis

Molecular structure analyses were performed using the ViennaRNA package [105], with minimum free energy (MFE) and spatial structure predictions made using the RNAfold Server (http://rna.tbi.univie.ac.at/, accessed on 6 January 2025). To quantify the reliability of the predicted structures, base pairing probabilities were calculated using a thermodynamic partition function, considering significant values > 0.7 (range: 0–1), which indicate robust conformational stability. In addition, the centroid structure that maximizes the expected pairing accuracy was generated, with statistical evaluation based on the probability distribution of the ensemble of secondary structures. Splicing variant alignments were conducted with the T-Coffee suite [106], which employs probabilistic consistency algorithms for global optimization. Each aligned residue received a CORE index (0–9), where scores ≥ 5 indicate statistically reliable alignments (*p* < 0.05 by sequence randomization). The consensus sequence was visualized and acquired in Jalview (v. 1.8.3) [107], with conservation scores calculated via normalized Shannon entropy (H(x) = −Σp(x)log_2_p(x)), considering values H < 0.2 as highly conserved. Consensus was established with a threshold of 40% identity per position. Phylogenetic reconstruction of the 30 transcripts was achieved using Jalview’s Neighbor-Joining algorithm applied to the T-Coffee alignment matrix, with evolutionary distances computed from neighbor-joining pairwise nucleotide p-distances (proportion of divergent sites, excluding gaps) and tree topology assessed via 1000 bootstrap replicates.

### 4.3. Interspecies Conservation Analysis

Inter-species conservation analyses were conducted using the Ensembl platform’s genomic comparison algorithm (https://www.ensembl.org/, accessed on 6 January 2025) [108], with metadata from GENCODE (https://www.gencodegenes.org/, accessed on 6 January 2025) [109], covering 91 vertebrate species. Refinement was performed using LncBook 2.0 (https://ngdc.cncb.ac.cn/lncbook/, accessed on 6 January 2025) [110] for 40 eutherian mammal species, based on UCSC genome alignments between humans and vertebrates. This process aimed to identify coding and non-coding homologous genes and determine the evolutionary age of human lncRNA genes. For gene representation, the best-aligned transcript was selected by comparing the isoforms, considering the one with the highest number of paired bases in the comparative analysis. The total alignment length was calculated as the sum of all aligned segments of the selected transcript, while sequence identity was determined by the weighted average (by length and identity) of all alignments. In the evaluation of sequence conservation, alignments with a minimum length of 50 nucleotides and coverage greater than 20% of the lncRNA transcript were considered homologous, requiring that the alignment performance (length and identity) exceed the Q50 threshold of introns—a parameter that represents the intermediate level of conservation observed in alignments of intronic regions. Illustratively, in the human-mouse comparison, transcripts such as TUG1 and MALAT1 were found to be exceptionally conserved, exceeding the Q99.5 threshold for introns. This approach allowed the identification of 139,306 homologous genes associated with 22,347 human lncRNA genes. The evolutionary age of lncRNAs was defined as the oldest phylogenetic period with homologous sequence occurrence, covering 17 hierarchical temporal nodes from the most recent to the most ancestral: ‘Homo’ (human-specific), ‘Hominini’, ‘Homininae’, ‘Hominidae’, ‘Hominoidea’, ‘Catarrhini’, ‘Simiiformes’, ‘Haplorrhini’, ‘Primates’, ‘Euarchontoglires’, ‘Boreoeutheria’, ‘Eutheria’, ‘Theria’, ‘Mammalia’, ‘Amniota’, ‘Tetrapoda’, and ‘Euteleostomi’, corresponding to key clades in the phylogenetic tree from zebrafish to humans.

### 4.4. Interactions Between LINC01133 and Tumorigenesis and Tumor Progression Pathways

Correlations between genes linked to tumorigenesis and tumor progression processes were analyzed using BioConductor packages (within the R Project for Statistical Computing Software 4.4.1 environment) [111] and the online suite cBioPortal for Cancer Genomics (Memorial Sloan Kettering Cancer Center, New York City, NY, USA, https://www.cbioportal.org/, accessed on 6 January 2025) [112]. Analyses were based on 11 pan-cancer studies (Cancer Therapy and Clonal Hematopoiesis (MSK, Nat Genet 2020); China Pan-cancer (OrigiMed, Nature 2022); MSK MetTropism (MSK, Cell 2021); MSK-CHORD (MSK, Nature 2024); MSK-IMPACT Clinical Sequencing Cohort (MSK, Nat Med 2017); MSS Mixed Solid Tumors (Broad/Dana-Farber, Nat Genet 2018); Metastatic Solid Cancers (UMich, Nature 2017); Pan-cancer analysis of whole genomes (ICGC/TCGA, Nature 2020); SUMMIT—Neratinib Basket Study (Multi-Institute, Nature 2018); TMB and Immunotherapy (MSK, Nat Genet 2019); and Tumors with TRK fusions (MSK, Clin Cancer Res 2020)) with an initial sample size (*n*) of 101,679 patient samples.

### 4.5. Methylation Profile in Tumors

The methylation levels of the LINC01133 gene in different tumor types were determined using the LncBook 2.0 platform (https://ngdc.cncb.ac.cn/, accessed on 8 January 2025) [110], provided by the China National Center for Bioinformation, Chinese Academy of Sciences. To characterize the DNA methylation profiles of lncRNAs in human diseases, LncBook 2.0 integrates 16 public bisulfite-seq datasets from TCGA (The Cancer Genome Atlas) and GEO (Gene Expression Omnibus), covering 14 types of cancer. The sets include data from acute lymphoblastic leukemia (GSE116229, 38 samples), acute myeloid leukemia (GSE135869, 15 samples), chronic lymphocytic leukemia (GSE113336, 18 samples), esophageal squamous cell carcinoma (GSE149608, 19 samples), medulloblastoma (GSE142241, 12 samples), and liver cancer (GSE79799, 6 samples), in addition to eight TCGA sets: bladder urothelial carcinoma (TCGA-BLCA, 7 samples), invasive breast carcinoma (TCGA-BRCA, 6 samples), colorectal adenocarcinoma (TCGA-COAD, 3 samples), lung adenocarcinoma (TCGA-LUAD, 6 samples), lung squamous cell carcinoma (TCGA-LUSC, 5 samples), rectal adenocarcinoma (TCGA-READ, 3 samples), gastric adenocarcinoma (TCGA-STAD, 5 samples), and uterine endometrioid carcinoma (TCGA-UCEC, 6 samples).

In identifying differentially methylated lncRNAs (characteristic lncRNA genes), promoter regions were defined as segments of −1500 bp relative to the transcription start site, calculating the average methylation level of all CpGs in the promoter or body regions. Considering the small sample size in some sets, specific criteria were applied: for ALL (GSE116229), AML (GSE135869), CLL (GSE113336), and ESCC (GSE149608), a maximum sample value > 0.2, median variation ≥ 2 or ≤1/2, and ad-justed *p*-value ≤ 0.05 (Wilcoxon test) was adopted; for MB (GSE142241) and RTT (GSE119980), a maximum value > 0.2 and *p*-value ≤ 0.02 (Wilcoxon); for the TCGA sets (BLCA, BRCA, COAD, LUAD, LUSC, READ, STAD, UCEC), a consistent in-crease/decrease in all case versus control samples was required, maximum value > 0.2 and control methylation level twice higher/lower than the maximum/minimum value of cases; for liver cancer (GSE79799), the minimum methylation level of cases/controls should exceed the maximum of controls/cases, with maximum value > 0.2 and median variation ≥ 4 or ≤1/4. Using these strict criteria, a total of 19,543 characteristic lncRNA genes with differential methylation profiles (hyper- or hypomethylated in the promoter or body region) were identified.

### 4.6. Narrative Literature Review

The narrative literature review was conducted by searching major scientific indexers (PubMed, https://pubmed.ncbi.nlm.nih.gov/, Google Scholar, https://scholar.google.com/, Web of Science, https://www.webofscience.com/, and Scopus, https://www.scopus.com/, accessed on 2 January 2025) for original articles, reviews, and systematic reviews published between 2015 and 2024. A total of 49 articles were selected, including 41 original research articles specifically on LINC01133 and tumors, seven articles on tumors and associated molecules (including LINC01133), and one review article specifically on LINC01133.

## 5. Conclusions

The study of lncRNAs in pathological contexts, although extremely recent, has already profoundly altered the paradigm of clinical and oncological therapeutic research. Their associations with various tumor-promoting and tumor-suppressing mechanisms open up a range of new possibilities that extend beyond emerging therapies and initiate a potential new era of treatments focused on ncRNAs.

LINC01133, like other lncRNAs of interest, such as HOTAIR, MALAT1, TUG1, and FOXCUT, should be at the forefront of clinical research, considering its importance both in the intratumoral context and in the tumor microenvironment.

The association of LINC01133 with modulation of cell proliferation pathways (Akt/PKB, Wnt/β-Catenin), which are related to regulation of the cell cycle, molecules associated with migration (E-cadherin, N-cadherin, Snail, Slug, Vimentin, Fibronectin), and invasion and metastatic processes (EZH2, Integrin, TNF-A, EGFR, VEGFR) provides a comprehensive outlook for future research, especially regarding its deep connection with pivotal molecules such as p53, p21, PTEN, c-Myc, and TGF-beta. Additionally, its role as an endogenous competitor, protein scaffold, and miRNA sponge is of utmost importance for more detailed investigations.

Thus, we believe that bioinformatics reviews and investigations such as the one presented here should encourage basic and clinical research on LINC01133 and provide periodic updates regarding its great potential in the oncological field.

## Figures and Tables

**Figure 1 ncrna-11-00058-f001:**
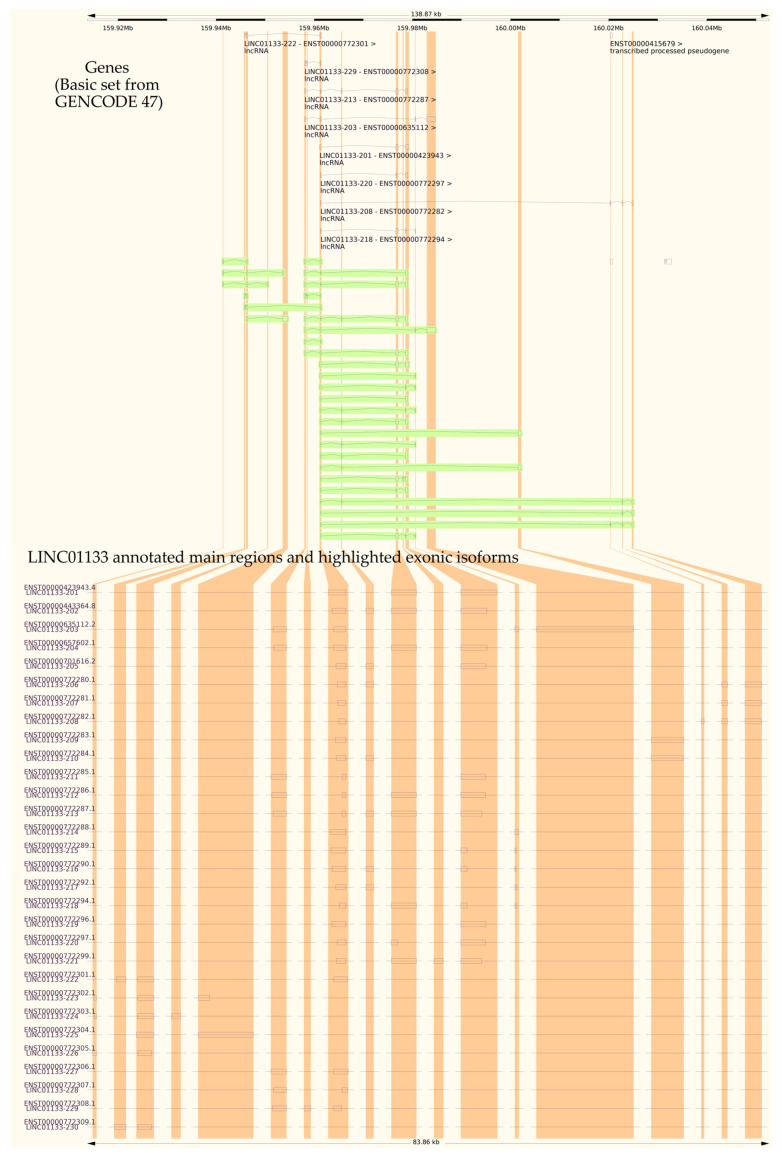
Association between the LINC01133 gene, its annotated transcripts, and their respective non-coding exons, obtained through the Ensembl and NCBI genomic and transcriptional annotation metadata.

**Figure 2 ncrna-11-00058-f002:**
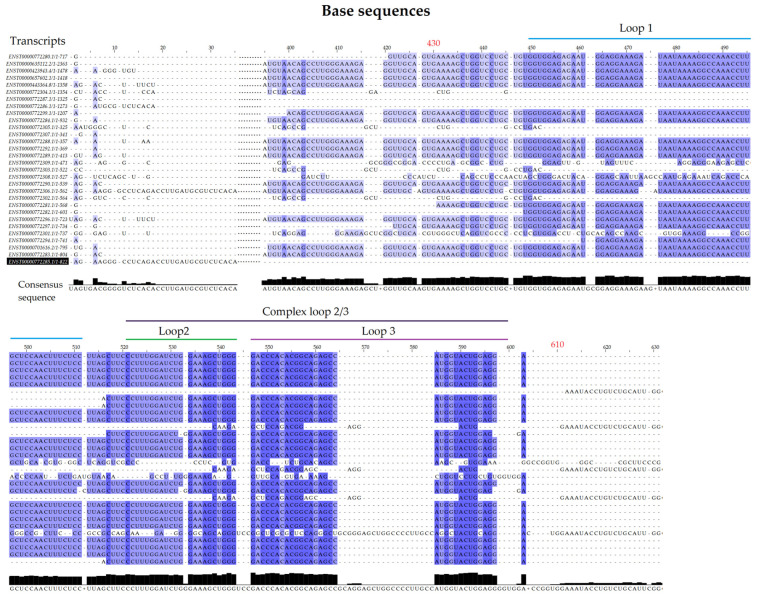
Alignment of the 30 existing LINC01133 transcript sequences using the T-Coffee suite WebServer coupled with JalView (v. 1.8.3) software, highlighting the regions between 1 and 600 bp and the consensus structure (between 430 and 610 bp, highlighted in red). The alignment was performed using global consistency methods via integration with ClustalW, and the consensus sequence was obtained via statistical analysis of weighted consensus probability.

**Figure 3 ncrna-11-00058-f003:**
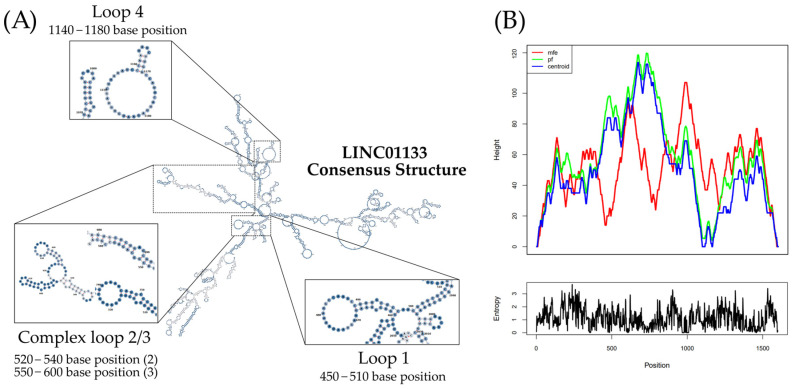
(**A**) Minimum free energy (MFE) flat structure of the predicted consensus structure of the lncRNA LINC01133. (**B**) Mountain plot representation of the thermodynamics of the predicted molecules and the positional entropy for each base position. Minimum free energy with MFE = −594.69 kcal/mol.

**Figure 4 ncrna-11-00058-f004:**
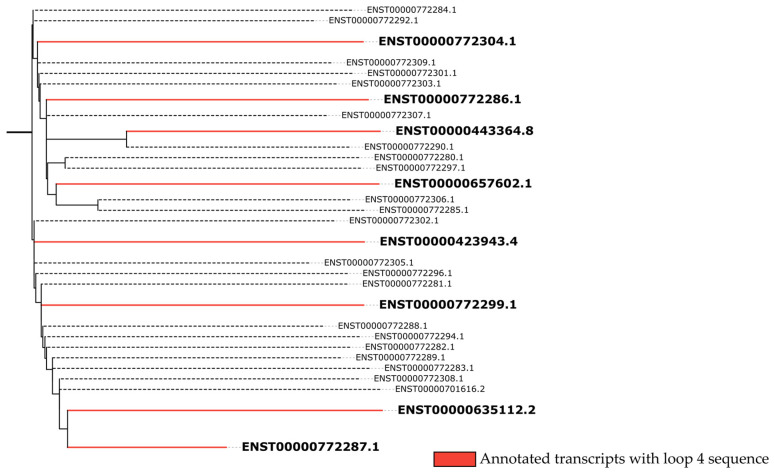
Conservation correlations between LINC01133 transcripts, with a focus (taxa/terminal groups in red) on transcripts containing internal loop 4. The analysis was performed via calculation of the matrix distance by pairwise identity, while the treeplot construction was carried out via Neighbor-Joining statistical analysis. (Significance threshold ≥ 5, *p*-value ≤ 0.05, confidence index 95%).

**Figure 5 ncrna-11-00058-f005:**
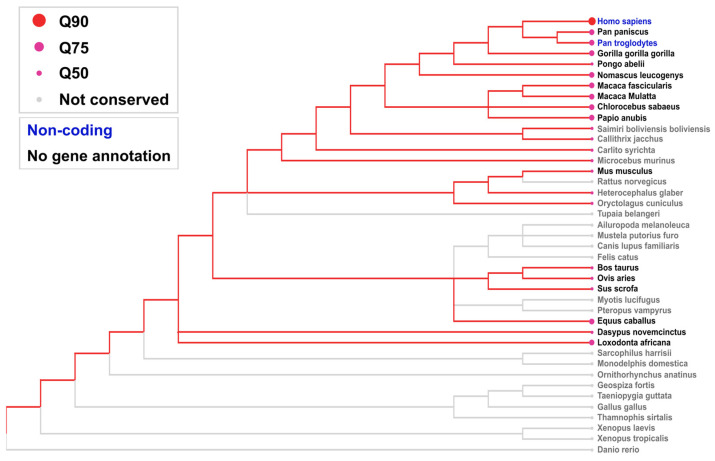
Gene conservation of LINC01133 from consensus transcriptional sequence among 41 vertebrate species. Conservation analyses were carried out through intronic alignments and correlated through similarity thresholds. Gray dots represent the absence of a gene conservation pattern or the absence of relevant data.

**Figure 6 ncrna-11-00058-f006:**
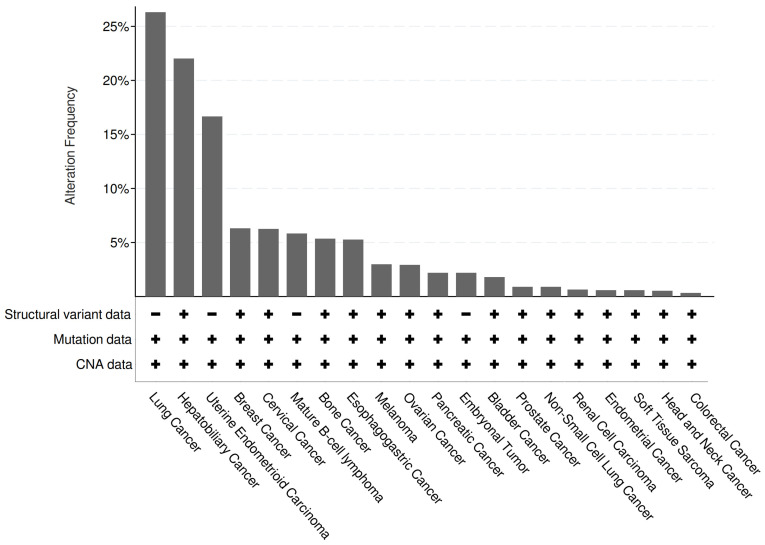
Genetic alterations of *LINC01133* in different types of tumors. The identification of structural variations was carried out via breakpoint detection; the Ensembl database was used to determine the associated mutations, searching for synonymous and non-synonymous mutations for categorization; the gene copy number alteration (CNA) data was analyzed via GISTIC.

**Figure 7 ncrna-11-00058-f007:**
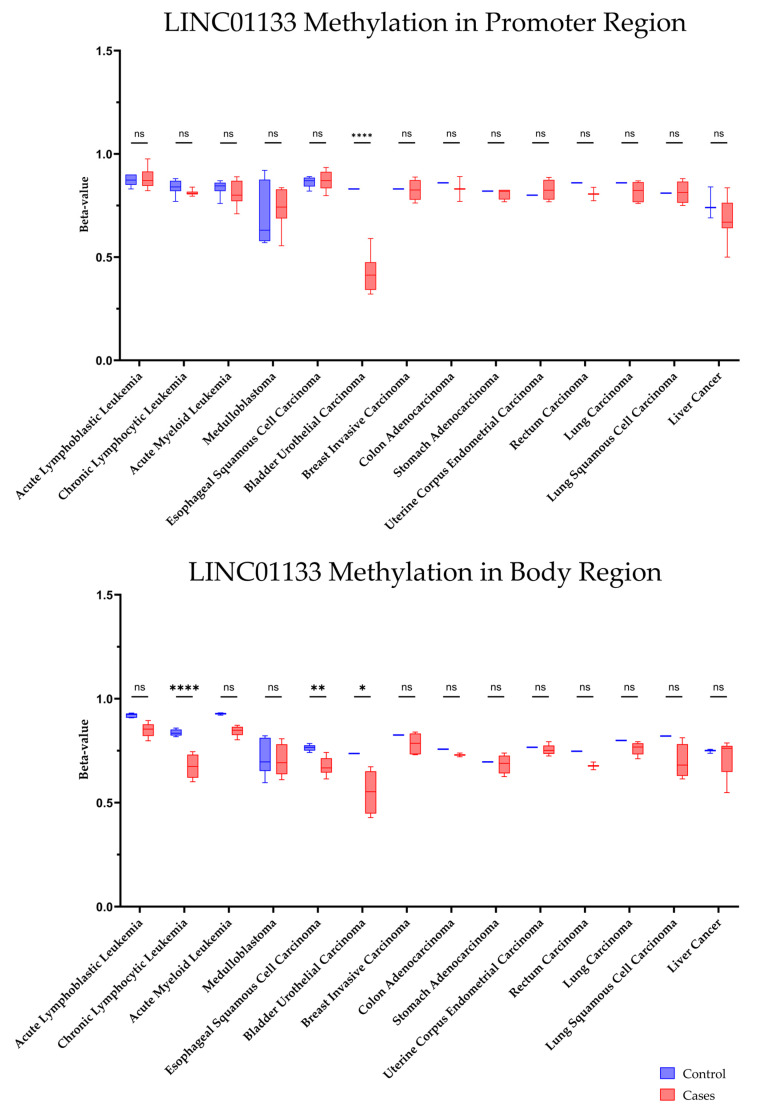
Differential methylation profiles of *LINC01133* in tumor types based on the comparison of methylation levels (beta-values) in the promoter and body regions of LINC01133 between tumor samples and controls, highlighting hypomethylation and hypermethylation patterns in 14 tumor types, with data processed via LncBook 2.0 from 16 bisulfite-seq sets (TCGA/GEO), with sample sizes ranging from 6 to 38 samples. Disease-specific statistical criteria were applied as described in the methodology (e.g., median variation ≥ 2x, maximum value > 0.2, directional consistency in TCGA samples); *p*-value (* ≤ 0.05; ** ≤ 0.01; **** ≤ 0.0001; 2-Way-ANOVA Test).

**Figure 8 ncrna-11-00058-f008:**
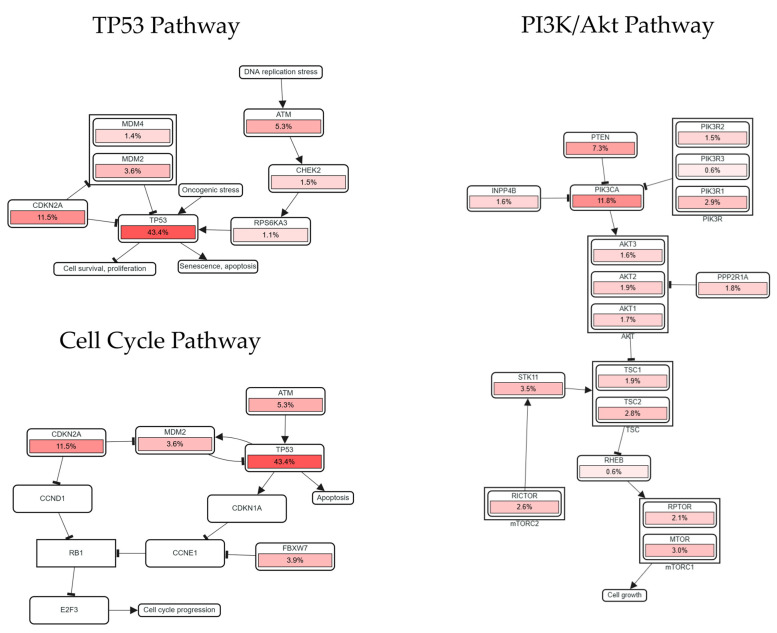
Positive correlations between aberrant expression of LINC01133 and key genes of pathways related to cell cycle destabilization and tumorigenesis. Analyses performed using Spearman’s correlation and FDR-adjusted significance (*q*-value < 0.05).

**Figure 9 ncrna-11-00058-f009:**
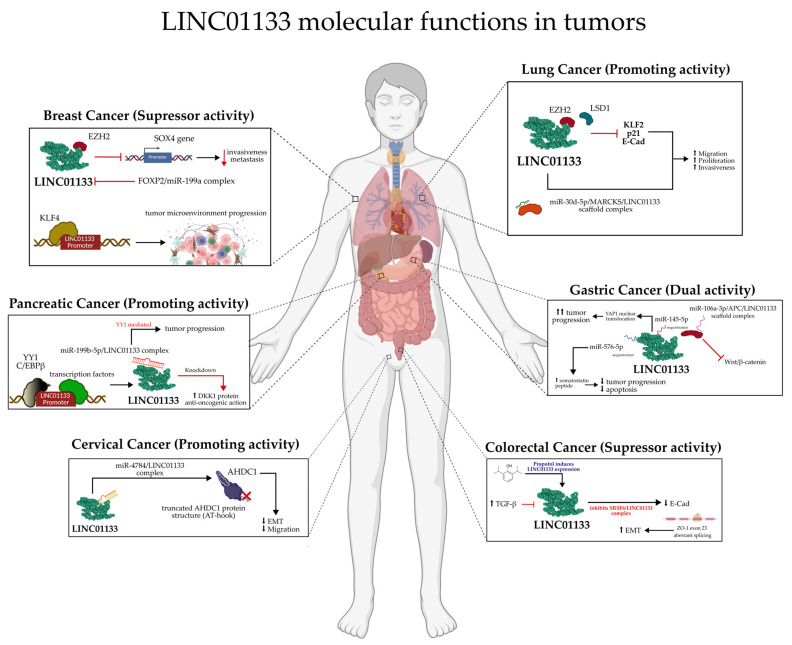
Key molecular mechanisms associated with LINC01133 in the major tumors with the highest incidence and mortality rates worldwide.

**Table 1 ncrna-11-00058-t001:** Annotated transcripts of LINC01133 and their length.

Transcript ID	Name	Base Pairs
ENST00000772280.1	LINC01133-206	717
ENST00000635112.2	LINC01133-203	2363
ENST00000423943.4	LINC01133-201	1478
ENST00000657602.1	LINC01133-204	1418
ENST00000443364.8	LINC01133-202	1358
ENST00000772304.1	LINC01133-225	1354
ENST00000772287.1	LINC01133-213	1325
ENST00000772286.1	LINC01133-212	1273
ENST00000772299.1	LINC01133-221	1207
ENST00000772284.1	LINC01133-210	932
ENST00000772285.1	LINC01133-211	822
ENST00000772283.1	LINC01133-209	804
ENST00000701616.2	LINC01133-205	795
ENST00000772294.1	LINC01133-218	741
ENST00000772301.1	LINC01133-222	737
ENST00000772297.1	LINC01133-220	734
ENST00000772296.1	LINC01133-219	723
ENST00000772282.1	LINC01133-208	601
ENST00000772281.1	LINC01133-207	568
ENST00000772302.1	LINC01133-223	564
ENST00000772306.1	LINC01133-227	562
ENST00000772290.1	LINC01133-216	539
ENST00000772308.1	LINC01133-229	527
ENST00000772303.1	LINC01133-224	522
ENST00000772309.1	LINC01133-230	471
ENST00000772289.1	LINC01133-215	413
ENST00000772292.1	LINC01133-217	369
ENST00000772288.1	LINC01133-214	357
ENST00000772307.1	LINC01133-228	341
ENST00000772305.1	LINC01133-226	325

**Table 2 ncrna-11-00058-t002:** Raw data on the methylation profile of the LINC01133 gene in tumors.

Source	Project ID	Disease Name (Short Name)	Sample Number
GEO	GSE116229	Acute Lymphoblastic Leukemia	38 (31 cases, 7 controls)
GEO	GSE135869	Acute Myeloid Leukemia	15 (9 cases, 6 controls)
GEO	GSE113336	Chronic Lymphocytic Leukemia	18 (11 cases, 7 controls)
GEO	GSE149608	Esophageal Squamous Cell Carcinoma	19 (10 cases, 9 controls)
GEO	GSE142241	Medulloblastoma	12 (8 cases, 4 controls)
GEO	GSE79799	Liver cancer	6 (3 cases, 3 controls)
TCGA	TCGA-BLCA	Bladder Urothelial Carcinoma	7 (6 cases, 1 control)
TCGA	TCGA-BRCA	Breast Invasive Carcinoma	6 (5 cases, 1 control)
TCGA	TCGA-COAD	Colon Adenocarcinoma	3 (2 cases, 1 control)
TCGA	TCGA-LUAD	Lung Adenocarcinoma	6 (5 cases, 1 control)
TCGA	TCGA-LUSC	Lung Squamous Cell Carcinoma	5 (4 cases, 1 control)
TCGA	TCGA-READ	Rectal Adenocarcinoma	3 (2 cases, 1 control)
TCGA	TCGA-STAD	Stomach Adenocarcinoma	5 (4 cases, 1 control)
TCGA	TCGA-UCEC	Uterine Endometrial Carcinoma	6 (5 cases, 1 control)

**Table 3 ncrna-11-00058-t003:** Main Pathways and Genes Associated with Tumorigenesis and Tumor Progression Processes.

Signaling Pathway	General Function	Main Genes and Components
WNT/β-Catenin	Regulation of cell proliferation, polarity, differentiation, and embryonic development. Its hyperactivation is associated with uncontrolled cell proliferation, resistance to apoptosis, and cell invasion [43].	CTNNB1 (β-catenin): main intracellular effector [44]. APC: tumor suppressor regulating β-catenin [45]. WNT ligands: activators of the pathway [46]. DKK and SFRPs: extracellular inhibitors of the pathway [47].
TP53	Regulation of genomic integrity by controlling the cell cycle and apoptosis in response to DNA damage. Its loss of function or mutation promotes genomic instability and oncogenesis [48].	*TP53*: central gene of the pathway [49]. MDM2 and MDM4: negative regulators of p53 [50]. ATM: kinases that activate p53 in response to DNA damage. CDKN1A (p21): inhibits cyclin-dependent kinases (CDKs) mediated by p53 [51].
PI3K/AKT	Regulation of cell survival, metabolism, angiogenesis, and proliferation. Its hyperactivation can result in resistance to apoptosis and tumor growth [52].	PIK3CA: catalytic subunit of PI3K frequently mutated in cancer. PTEN: tumor suppressor antagonizing the pathway. AKT1/2/3: intracellular signaling mediators. MTOR: regulates protein synthesis and metabolism [52].
NRF2	Control of the antioxidant response by regulating genes that protect cells from oxidative stress. Its overexpression promotes adaptation to the stressful tumor microenvironment [53].	NFE2L2 (NRF2): main transcription factor. KEAP1: negatively regulates NRF2, promoting its degradation. CUL3: component of the ubiquitination complex [53].
TGF-β	Functional duality: in early stages of cancer, it acts as a tumor suppressor; in advanced stages, it promotes invasion, metastasis, and tumor microenvironment remodeling [54,55].	TGFB1-3: pathway ligands. TGFBR1/2: receptors initiating signaling. SMAD2/3: intracellular mediators that translocate to the nucleus. SMAD4: coactivator of SMADs [54].
MYC	Regulation of gene transcription linked to cell growth, metabolism, ribosomal biogenesis, cell cycle, and apoptosis [56].	MYC (c-Myc): central transcription factor. MAX: forms heterodimers with Myc to activate transcription. MXD Complex: interacts with Myc to suppress proliferative genes [56].
RTK-RAS	Transduction of extracellular signals for controlling cell proliferation, survival, and differentiation. Overexpression of the pathway, especially in solid tumors [57].	EGFR, ERBB2 (HER2): frequently amplified tyrosine kinase receptors. KRAS, NRAS, and HRAS: members of the Ras family frequently mutated in cancer. RAF1/BRAF: downstream effectors activated by Ras [57].
NOTCH	Control of cell fate decisions during development and homeostasis. Its deregulation leads to abnormal proliferation and resistance to apoptosis [58].	NOTCH1-4: pathway receptors. JAG1/2: ligands that activate Notch [58]. HES/HEY: target genes regulating cell differentiation [59].
Cell Cycle	Coordination of cell cycle progression and control. Alterations in checkpoint controls lead to deregulated cell division, a hallmark of tumors [48].	FBXW7: inhibits CDK activators. RB1: regulates the cell cycle restriction point. TP53 and CDKN2A: controls the response to DNA damage and cell cycle arrest [48,49].
HIPPO	Regulation of organ size, cell proliferation, and apoptosis. Its inactivation promotes tumor growth and resistance to apoptosis [60].	LATS1/2: regulators that inhibit downstream effectors. YAP/TAZ: transcriptional effectors promoting cell growth. TEAD: transcription factor activated by YAP/TAZ. NF2 (Merlin): upstream tumor suppressor [60].

**Table 4 ncrna-11-00058-t004:** Association between LINC01133 expression and its action in tumors.

Tumor	Studies	Promoting (P), Suppressing (S) or Both (B) When Expression Is Significant	References
Lung	5	(P) in LSCC (P) in NSCLC (P) in LUSC	[63,64,65,66,67]
Colorectal	3	(S) in CRC	[68,69,70]
Breast	3	(S) in TNBC (S) in Luminal	[71,72,73]
Ovarian	2	(B) in EOC * * (1 study in cell lines 1 study in patients’ ovarian samples)	[74,75]
Bladder	1	(S) in BC	[76]
Hepatocellular	2	(P) in HCC	[77,78]
Pancreas	10	(P) in PC (P) in PAAD (P) in PDAC	[79,80,81,82,83,84,85,86,87,88]
Esophagus	1	(S) in ESCC	[89]
Oral Squamous Cells	1	(S) in OSCC	[90]
Gastric	5	(B) in GT * * (1 study-(P))	[91,92,93,94,95]
Nasopharynx	1	(S) in NPC	[96]
Cervical	3	(P) in CC (P) in CESC	[97,98,99]
Renal	2	(P) in RCC	[100,101]
Bone	1	(P) in BC	[102]
Endometrium	1	(P) in EC	[103]

## Data Availability

Data supporting the findings of this study are available upon reasonable request from the corresponding author.

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
