# Peer review of "The Good, the Bad, or Both? Unveiling the Molecular Functions of LINC01133 in Tumors"

_ncrna, 2025, doi:10.3390/ncrna11040058_

Round 1

Reviewer 1 Report

Comments and Suggestions for Authors
  1. The Figures included in the paper need to be sized properly. In the present form, the images are too small, and when increased, they are blurry. It is ok for the Figures to require some zooming in for the reader to better observe some details, but the image on 100% of paper size needs to be visible enough for the reader to understand the main message. Apart from Figure 6, the size and the fonts of all other images need to be increased, and also the quality should be enhanced as they appear rather pale in the present form.
  2. Figure 2 legend makes a reference to Figure 3 for internal loops structure. It is uncommon to reference a figure from later in the text, so I suggest these figures are merged into one figure with A and B panels.
  3. Figure 3 - I am unable to zoom in properly on the image, but from what is visible in the present form, the red circles, that I am assuming are part of the secondary structure of LINC01133, seem to be disconnected to rest of the molecule (is looks as if they are randomly positioned on top of green structures). Please improve the quality of the image, as already stated in comment 1,  and ensure the structure presentation is correct. Also, add an explanation in the figure legend for the area squared in red.
  4. The results presented in Figure 4 were not explained in detail in Material and Methods section. I was assuming this stems from interspecies conservation analysis, but the methods mentioned in the Figure 4 legend do not match the methods stated in that section of M&M.
  5. Li 270 mentions "gray plots", I assume this explanation belongs to figure legends; the text should just describe the results. 
  6. Li 284-6, while the statements here and the references are correct, it’s not entirely clear why they are mentioned in the context of conservation, so I suggest a better explanation. 
  7. Li 303 - which evidence points out to this? It should be more clearly elaborated, irrespective of the low percentage of samples where alterations are detected, the type of variation is not enough to simply say a gene is a tumor suppressor.
  8. Figure 8 is not referenced in the main text, and , as the other figures, is not legible. Therefore it is not clear where the results described in li 361 - 373 stem from.

Author Response

Please, see the attachment. Thanks!

Reviewer 2 Report

Comments and Suggestions for Authors

In this review, Júnior et al provided computational analysis of sequence and structural features of LINC01133 and literature review of current evidence of its roles in cancers. While the topic presented is interesting, I do have several comments on significant edits of the figures and text, mostly regarding the computational analysis included in the review (I do like the literature review section and believe that those are of great value):

  1. For most of the figures: I can hardly read the text or any labels from the figures. Please consider remake the figures or add readable labels.
  2. Figure 3: In the RNAfold prediction, red means high base-pairing probability. However, the red sequences in the structures are apparently unpaired. Can the authors explain why? Or if the authors actually mean sequences with inter-RNA base-pairing probability, please explain how such predictions are made in the text. In the figure legend, please also explain what each number and the red box represent (I assume those are the internal loops).

It's also not crystal clear to me what sequences are used for the prediction. It seems that a ‘conserved’ sequence is used. But it’s not obvious to me that the sequences in the red box are conserved, according to Figure 2.

To show structural similarities between different isoforms, I believe structural predictions should be performed for each isoform separately, followed by comparison of the predicted structures.

  1. Figure 5: I don’t see any species with the annotation of LINC01133 as protein coding, and in most species, there is no associated gene annotation. Is this a mislabeling?
  2. Relevant to Figure 5: the authors mentioned LINC01133 having 5'and 3'UTR, which are RNA regions well defined for mRNAs but not ncRNAs, unless it has coding potentials. The protein coding potential of LINC01133 needs to be discussed early to avoid confusion, if any.
  3. Page 9: “While insufficient data precludes the establishment of precise hypotheses regarding 301 the convergent function of LINC01133 in tumors, evidence suggests its strong association 302 as a tumor suppressor.”

Can the authors briefly discuss why this is so? What is the evidence?

  1. Figure 7: please add p values from statistical test.
  2. Figure 9: a different structure on LINC01133 is presented, which isoform is this from?

Author Response

Please, see the attachment, thanks!

Reviewer 3 Report

Comments and Suggestions for Authors

Major comments:

  1. Quite a substantial improvement is needed in terms of figure presentation, most of the figures are really difficult to read.
  2. Much more detail is needed on the background around this LINC RNA, its discovery, its structure, the nature of its isoforms, and its normal expression profile, etc
  3. The mechanisms around its involvement in cancer need to be explained in much greater detail. Less on correlations, more needed on mechanism. To illustrate this point, for example, take lines 408-414. LINC01133 interacts with EZH2/LSD1 but how? A series of techniques is listed in line 411 - no detail is provided in terms of what exactly the experiments found. So in revising this manuscript I would recommend providing a lot more detail and explanation around the processes illustrated in Figure 9.
  4. Much of the article comes across as a list, with some sections given much more prominence. Many of the sections on individual cancers are rather short.
  5. The title points to the proposed dual nature of the LINC RNA, but this particular theme (its duality) is a little bit lost in the article. A table that perfectly summarises the evidence of this duality would be beneficial.

Comments:

  1. The Abstract needs to be rewritten. There are a number of issues. The quality of English needs to be improved. What is meant by "capped" in line 12? Line 22, how do structural motifs bind to signaling pathways? line 28 a sentence can't start with Could be, and why is Cancer in caps? etc
  2. Avoid single-sentence paragraphs, eg line 35, line 52 etc. and many others.
  3. Line 47 the concept of multifunctionality is not new; and has applied to many proteins for several decades now. This could be elaborated.
  4. I wonder if a Figure might help illustrate the four types of lncRNAs described, line 60-65.
  5. The introduction could include some information on the regulation of expression of lncRNA, including their alternative splicing.
  6. Methods, is it really necessary to describe the "narrative literature review"?
  7. Line 203 what is the nature of the two variants? describe these. Similarly, it is not enough just to list the many transcripts, the difference in their structure needs to be included. Figure 2 does not provide this interpretation, and is very difficult to read.
  8. Figure 1 is also very difficult to read and interpret. In fact in general, the figures need enhancement, eg Figure 7 is really crowded and difficult to read.
  9. Other than the predictions of structure through software, is there any physical evidence in the literature that these potentially conserved loops exist and are functionally important?
  10. I don't entirely follow the logic around sequence conservation - suggesting strong correlation specifically with tumorigenic processes? unless the conserved feature is one that has previously been shown to be important in tumorigenesis.
  11. Nor is the link to TP53 copy number necessarily clear - highly speculative.
  12. When directly mentioning genes (eg TP53) italicise.
  13. The final statement lines 301-303, is contradictory, and not backed up by sufficient evidence - this section on RNA-Seq is very brief and there is little there.
  14. Not entirely sure of the point of table 3, specifically how all these processes relate to LINC01133?
  15. Sections on bladder, esophageal, nasopharyngeal, bone, renal and endometrial tumours are rather short.
Comments on the Quality of English Language

There are some relatively minor but consistent issues with the quality of English which would need to be addressed at revision stage.

Author Response

Please see the attachment, thanks!

Round 2

Reviewer 1 Report

Comments and Suggestions for Authors

I thank the authors for addressing all of my concerns. The manuscript is now improved, especially given that the figures have been significantly redesigned. I will just comment on two points from the previous review:

  1. Li 270 mentions "gray plots", I assume this explanation belongs to figure legends; the text should just describe the results.

Response 5: We removed all mentions of grayplots from the main text and incorporated them into the caption of Figure 5. Additionally, we readjusted the data within the figure, focusing only on species exhibiting significant conservation of LINC01133 relative to Homo sapiens

Reviewer’s response: The image size is correct now, I just don’t see any of the data being readjusted. Given that I did not specifically suggest the data to be changed, I am considering this comment to be resolved.

  1. Li 303 - which evidence points out to this? It should be more clearly elaborated, irrespective of the low percentage of samples where alterations are detected, the type of variation is not enough to simply say a gene is a tumor suppressor.

Response 7: The role of LINC01133 as a tumor suppressor across various tumor types is presented in the narrative literature review concluding the manuscript. We agree that the data in Figure 6 may establish correlations for some, but not most, tumor types. Furthermore, its oncogenic or suppressive role cannot be inferred solely from the indicated alteration frequency. Therefore, this section of the manuscript has been entirely rewritten to clarify these aspects.

Reviewer’s response: I appreciate the authors rewriting this passage, I would only suggest adding references to statements that refer to data from the literature.

Reviewer 2 Report

Comments and Suggestions for Authors

The authors have made significant changes to the text and figures to address my concerns and questions - I do not have further comments and find the new text suitable for publication.

Reviewer 3 Report

Comments and Suggestions for Authors

I would like to thank the authors for addressing all of my points in detail.

I do still feel that figure clarity could be improved, not the content, but the use of larger fonts because I still find it hard to read some of them.